# Polycaprolactone-Based 3D-Printed Scaffolds as Potential Implant Materials for Tendon-Defect Repair

**DOI:** 10.3390/jfb13040160

**Published:** 2022-09-23

**Authors:** Merle Kempfert, Elmar Willbold, Sebastian Loewner, Cornelia Blume, Johannes Pitts, Henning Menzel, Yvonne Roger, Andrea Hoffmann, Nina Angrisani, Janin Reifenrath

**Affiliations:** 1Hannover Medical School, Clinic for Orthopaedic Surgery, Anna-Von-Borries-Str. 1–7, 30625 Hannover, Germany; 2German Cancer Research Center, Center for Preclinical Research, Im Neuenheimer Feld 280, 69120 Heidelberg, Germany; 3Hannover Medical School, Lower Saxony Centre for Biomedical Engineering, Implant Research and Development (NIFE), Stadtfelddamm 34, 30625 Hannover, Germany; 4Institute of Technical Chemistry, Leibniz University Hannover, Callinstraße 5, 30167 Hannover, Germany; 5Institute for Technical Chemistry, Braunschweig University of Technology, Hagenring 30, 38106 Braunschweig, Germany

**Keywords:** printing, degradable, defect repair, tendon graft, cytocompatibility, surface modification

## Abstract

Chronic tendon ruptures are common disorders in orthopedics. The conventional surgical methods used to treat them often require the support of implants. Due to the non-availability of suitable materials, 3D-printed polycaprolactone (PCL) scaffolds were designed from two different starting materials as suitable candidates for tendon-implant applications. For the characterization, mechanical testing was performed. To increase their biocompatibility, the PCL-scaffolds were plasma-treated and coated with fibronectin and collagen I. Cytocompatibility testing was performed using L929 mouse fibroblasts and human-bone-marrow-derived mesenchymal stem cells. The mechanical testing showed that the design adaptions enhanced the mechanical stability. Cell attachment was increased in the plasma-treated specimens compared to the control specimens, although not significantly, in the viability tests. Coating with fibronectin significantly increased the cellular viability compared to the untreated controls. Collagen I treatment showed an increasing trend. The desired cell alignment and spread between the pores of the construct was most prominent on the collagen-I-coated specimens. In conclusion, 3D-printed scaffolds are possible candidates for the development of tendon implants. Enhanced cytocompatibility was achieved through surface modifications. Although adaptions in mechanical strength still require alterations in order to be applied to human-tendon ruptures, we are optimistic that a suitable implant can be designed.

## 1. Introduction

Tendon disorders are one of the major diseases in orthopedics [1], which often lead to functional limitations and long-term pain. In addition to acute trauma, intrinsic factors such as age and health status play a role in the pathogenesis, as do extrinsic factors, such as smoking and nutrition [2]. If these tendon injuries are not treated immediately or adequately, long-term damage, such as fatty degeneration, extensive retraction and the formation of scar tissue may occur [3]. Especially for ruptures of large tendons, surgical treatment is often unavoidable. However, due to the lack of suitable implant materials to support conventional surgical methods, re-ruptures are common complications, with rates of between 11% and 94% [4]. Even if a re-rupture does not occur, the resulting reparation tissue is often insufficient due to its lower load-bearing capacity [5]. To avoid these high rates of re-rupture and loss of functional tissue, it is of greatest interest to develop implant materials with good biological and mechanical properties to support tissue regeneration after surgical-tendon refixation.

Polycaprolactone (PCL) is a promising candidate through which to produce such scaffolds. PCL is a biodegradable polyester with a degradation rate of two to four years. It degrades to slightly acidic metabolites [6]. It offers the opportunity for the body to replace the scaffold with functional tissue without long-term residues. PCL is able to absorb higher forces and shows higher stiffness compared to other polymers, such as polyglycolic acid (PGA), poly-l-lactid acid (PLLA) or their copolymers (PLGA) [7,8]. The elasticity of PCL is mainly dependent on the molecular weight [9]. PCL is already used as a suture material or as a carrier for drug-delivery systems [6]. In the literature, it has been studied as a tendon-implant material with different electrospun or woven design variants [10,11,12,13], but currently, there is no commercially available PCL construct for tendon ruptures [14]. As a FDA-approved material, PCL induces, for example, massive foreign-body reactions in vivo when used as an electrospun fiber mat with a very small fiber diameter [15]. Therefore, more research is needed to produce varied designs. In addition, its cost-effectiveness and high durability make it interesting for commercial production [16]. Electrospun PCL was evaluated in preclinical animal models for chronic tendon defects, but it did not convince the researchers due to its limited mechanical strengths and the foreign-body reactions observed [17].

Three-dimensional-printing technology offers the possibility of processing PCL in a more controlled manner than with electrospinning and of adapting the mechanical and biological parameters [18]. Using computer-aided design (CAD) software, scaffolds can be produced with highly controllable properties, such as size, porosity and interconnectivity [19,20]. The ease of handling also makes it very attractive for use in conventional settings.

In the process “from bench to bedside”, biocompatibility is an important aspect. A known problem in the use of PCL is its high hydrophobicity and poor wettability, which are unattractive for cells and result in minor cell growth on the material’s surface [21]. For this reason, surface modifications to PCL are of great interest in the development of PCL-based implants. Potential options through which to increase the wettability include plasma-chemical methods, which create functional groups on the polymer surface and thereby increase the hydrophilicity [22]. Depending on the applied gas, different functional groups are formed. In the case of oxygen low-pressure plasma treatment, carbonyl, carboxyl or hydroxyl groups are formed [23]. Another good possibility for surface modification is offered by biomimetic coatings with special proteins. In this method, proteins which are part of the extracellular matrix and form integrins for the formation of cell contacts are suitable candidates. These proteins include collagen, fibronectin and laminin [24,25]. Due to the fact that polymers are almost inert due to their aliphatic CH structure, a combination of these processes is often used in order to achieve the best possible coating success [26].

The aim of this study was to fabricate, characterize and modify 3D-printed PCL-based scaffolds with adequate mechanical and biological properties for possible later use in tendon-implant applications. The authors’ main hypotheses were that 3D-printed PCL based specimens would increase load-bearing capacities compared with electrospun specimens, that cell adhesion in contact with the printed and surface-treated material would be possible, and that the cell bodies would be spindle-shaped in its appearance, indicative of cytocompatibility.

## 2. Materials and Methods

### 2.1. Bulk Material

Two different commercially available PCLs were used as bulk materials. (Table 1). Molecular weight was characterized at the Technical University Braunschweig. Gel permeation chromatography (GPC) measurements were performed on a PSS SECcurity2 instrument equipped with a SECcurity2 vacuum degasser, a SECcurity2 TCC6500 column oven (all PSS GmbH, Mainz, Germany), a 1200 isocratic pump (Agilent Technologies Germany GmbH & Co. KG, Waldbronn, Germany), a SECcurity2 1200 refractive index detector (PSS GmbH, Mainz, Germany) and a manual injection valve (Rheodyne^®^, IDEX Health & Science LLC, Oak Harbor, WA, USA). Both PCL species were recorded in THF at 40 °C on a PSS triple SDV column setup (1 × pre-column, 2 × main columns, each 10 µm particle and pore size) with a concentration and flow rate of 1 g/L and 1 mL/min, respectively.

### 2.2. Manufacturing of Test Specimens

The commercially available software, autoCAD^®^ (Autodesk, San Raphael, CA, USA), was used to design different test specimens of 3D-printed PCL scaffolds.

An extrusion -based BioX Cellink bioprinter (Cellink, Freiburg, Germany) was used for specimen printing (Figure 1A). PCL granules were melted in the printing cartridge, melted PCL was pressed through a 150-micrometer nozzle and specifically placed on a glass slide in three layers. Printing of PCL45 was carried out at 80 °C, 700 kPa and 8 mm/s, PCL86 was printed at 135 °C, 700 kPa and 3 mm/s. Two different designs were chosen: design 1, with a square pattern; and design 2, with a diagonal interlayer between the crossed layers (Figure 1B).

For mechanical testing, specimens according to DIN EN 5272 (Figure 2A) with 0.5-millimeter thickness were designed. For further-cell culture tests, specimens with dimensions of 10 mm × 10 mm × 0.5 mm were produced.

For characterization of implants, microscopic pictures (25× magnification, Figure 1C) were analyzed and mean filament diameter was calculated from 20 randomly selected measurement points per mat. PCL86 specimens showed a mean filament diameter of 174.21 ± 28.50 µm and PCL45 specimens a mean filament diameter of 259.24 ± 13.74 µm.

Additionally, test specimens of electrospun fiber mats were prepared by cutting into test-specimen shape as reference material to compare the printed specimens with formerly used electrospun material from in vitro and in vivo studies [17,27]. Manufacturing of these reference materials is described by de Cassan et al. [28] and Fricke and Becker [29].

Test specimens were fixed between two clamps in a material test machine (Zwick 1445, Ulm, Germany), as shown in Figure 2B. Preload was applied (0.5 N, 5 s) and further preconditioning with 7 cycles and 3% elongation at a frequency of 1 Hz was carried out. Finally, specimens were pulled with a speed of 20 mm/min until failure and force-distance diagrams were recorded. Stiffness (N/cm) was calculated as gradient of the curve in the linear region. Elongation at break was calculated as quotient of initial length (L0 (mm)) and length at failure (LFmax (mm)).

### 2.3. Surface Hydrophilization with Oxygen-Plasma Treatment

For treatment with oxygen plasma, a MiniFlecto^®^ plasma oven (Plasma Technology, Herrenberg, Germany) was used. Different values of treatment parameters (time, pressure and energy) were tested (n = 3 per group, Table 2). Hydrophilization success was examined by contact-angle measurement in accordance with the “sessile-drop” method [30]. A total of 20 µL distilled water was applied and two contact angles were measured per specimen with a contact-angle-measurement device (OCA200, DataPhysics Instruments GmbH, Filderstadt, Germany).

Prior to further use, specimens were shrink-wrapped in sterilization pouches (Henry Schein, Melville, NY, USA) and subjected to electron-beam sterilization with a Rhodotron TT100 e-beam accelerator at 25 kGy (Mediscan, Kremsmünster, Austria).

### 2.4. Coating with Fibronectin and Collagen I

Plasma-treated specimens (treatment f) were further coated with different coating procedures (fibronectin or collagen I) and therefore placed in 24-well plates.

Human fibronectin (Corning, Glendale, CA, USA) was diluted in phosphate buffered saline (PBS) to a concentration of 25 µg/mL. A total of 200 µL of this solution was applied to each specimen and incubated for 1 h at room temperature. After washing in PBS, specimens were directly used for cell-culture tests.

Collagen Type I rat tail 10 mg/mL (Ibidi, Gräfelfink, Germany) was diluted with 17.5 mM acetic acid to a ready-to-use solution of 50 µg/mL and stored at −20 °C until usage. For collagen coating, specimens were incubated in 400 µL collagen Type I in acetic acid solution for 1 h at room temperature. After washing in PBS, specimens were directly used for cell-culture tests.

### 2.5. Cell-Culture Tests

It was only possible to obtain PCL86 at a level of purity suitable for medical applications. Therefore, further experiments were conducted only on PCL86. In a first step, L929 fibroblasts were cultured on PCL86 specimens with two different design variants (design 1 (PCL86_1) and design 2 (PCL86_2), Figure 1).

For testing with L929-fibroblasts, approximately 1.2 × 10^6^ cells were thawed and cultured in a T175 cell-culture flask. Incubation was performed in DMEM (FG0415, Biochrom, Berlin, Germany) plus supplements (10% FCS (Thermo Fischer Scientific, Waltham, MA, USA), 1% penicillin–streptomycin (Capricorn Scientific GmbH, Ebsdorfergrund, Germany) and 1% Glutamax (Thermo Fischer Scientific, Waltham, MA, USA) at 37 °C and 5% CO_2_ for up to 70–80% of confluence. Cells were detached from the culture flasks with trypsin-EDTA (Pan-Biotech GmbH, Aidenbach, Germany) and counted using a Casy cell counter (CASY, OMNI Life Science GmbH, Basel, Switzerland). Sterile PCL86-specimens (d1 and d2, Table 3) were placed in a 24-well plate and seeded with approximately 20,000 cells per well in 500 µL DMEM plus supplements. As positive control, cells were seeded in empty wells (n = 6). Subsequently, cell-culture plates were placed in an incubator at 37 °C and 5% CO_2_ for up to 7 days.

Cell viability was tested after 24 h, 3 d and 7 d using sterile WST-8 tests (Colorimetric cell viability kit I, PromoCell GmbH, Heidelberg, Germany). Cellular dehydrogenases transformed the tetrazolium-salt WST-8, depending on cell viability, to the orange-colored WST-8 formazan. In a first step, cell-culture medium was replaced by 200 µL fresh medium (DMEM plus supplements). A total of 20 µL of WST-8 solution (1:10) was added in each well and incubated for 4 h at 37 °C and 5% CO_2_. A 100-microliter supernatant of each well was used for colorimetric measurement in a microplate reader at 450 nm absorption. In case of signals higher than the measurable range, supernatant was diluted accordingly (1:1 in PBS).

In a second step, human-bone-marrow-derived mesenchymal stem cells (hbm-MSCs) were used for further in vitro characterization (ethical vote #565 to Andrea Hoffmann by institutional ethics committee). The 24-well-plates were coated with polyHEMA to avoid cell attachment and growth at the bottom of the plates. One g polyHEMA (Sigma, Darmstadt, Germany) was diluted in 50 mL 1:1 acetone/ ethanol mixture (both Sigma, Darmstadt, Germany), vortexed and panned until complete dissolution of polyHEMA. Sterile filtration was performed with a PVDF-sterile-filter (0.22 µm) and 100 µL were inserted in each well for 5 min. Subsequently, plates were dried and washed twice with PBS.

Hbm-MSCs were thawed and cultured in a T175 cell-culture flask with a density of at least 2000 cells per cm^2^ in DMEM (FG0415, Biochrom, Berlin, Germany) supplemented with 10% FCS (Thermo Fischer Scientific, Waltham, MA, USA), 25 mM 4-(2-hydroxyethyl)-1-piperazineethanesulfonic acid (HEPES) buffered solution (Sigma, Darmstadt, Germany), 100 U/mL penicillin/100 μg/mL streptomycin (Capricorn Scientific GmbH, Ebsdorfergrund, Germany) and 2 ng/mL recombinant human FGF-2, (Peprotech, Rocky Hill, CT, USA) for 2–3 d to a confluence of 70–80%. Cells were detached and counted as described for the L929 cells.

Specimens (n = 6 per group, Table 3) were placed in polyHEMA-coated 24-well plates. Each specimen was seeded with 20,000 cells in 500 µL of the respective medium and placed in the incubator at 37 °C and 5% CO_2_ until further processing.

Cell viability was tested after 24 h, 3 d and 7 d, as described for L929 cells.

For the evaluation of cell morphology and cell attachment on the 3D-printed specimens, exemplary specimens were removed at each time point (n = 1 after 24 h; n = 1 on day 3; n = 4 on day 7), fixated in 4% buffered paraformaldehyde for 20 min, washed twice in PBS and treated with 0.1% Triton X-100 (Carl Roth GmbH & Co.KG, Karlsruhe, Germany). After a further washing step, incubation in 0.3 µM Phalloidin-TRITC und 0.1 µg/mL DAPI (both Sigma, Darmstadt, Germany) for 60 min was performed. Microscopic images were taken at 100× magnification using a confocal laser-scanning microscope (CLSM, Leica Microsystems, Wetzlar, Germany) and representative areas were assessed for cell distribution and morphology.

### 2.6. Statistics

Mean values and standard deviation were determined for all parameters. Differences between treatment procedures (plasma treatment, coating) were analyzed with ANOVA (SPSS version 27, IBM Deutschland gmbH, Ehningen, Germany) for normally distributed, independent data. Post hoc test was performed with Tukey when variance equality was assumed or Games–Howell if assumption of variance equality was not fulfilled. In case of non-parametric data or abnormal distributed data, the Kruskal–Wallis test was used with *p* < 0.05 as the threshold for significance in all statistical analyses.

## 3. Results

### 3.1. Mechanical Examination of Specimens

Compared to the previously used electrospun specimens with a maximum force at failure of 2.19 ± 0.80 N, all the 3D-printed materials showed significant higher values. The highest forces at failure (12.69 ± 1.48 N) were reached with the PCL45. Similar results were obtained for the stiffness. The PCL45 was significantly stiffer than the PCL86 and the electrospun PCL showed significantly lower values compared to all the 3D-printed materials. The elongation at break was highest in the electrospun specimens and similar in the 3D-printed materials (Figure 3).

Different printing patterns of PCL86 showed a trend (*p* = 0.086) towards higher forces in the specimens with diagonal middle layers (8.51 ± 1.96 N, d2) compared to square patterns (6.28 ± 1.60 N, d1). No differences were detected between the designs in terms of stiffness and elongation.

### 3.2. Influence of Design Variant and Plasma Treatment on Hydrophilicity of Specimens

The specimens after plasma treatments d, g, i and k showed macroscopic changes directly after the treatment procedure, including fusion, color changes and shrinking. They were excluded from further evaluation.

The contact-angle measurements showed increases in surface hydrophilization for all the other treatment procedures compared to the native controls. The contact angles of the native specimens were 110.63 ± 1.37° (PCL86_1) and 89.82 ± 1.22° (PCL86_2); they were significantly reduced to 16.32 ± 1.27° (PCL86_1_f, *p* = 0.08) and by trend 26.45 ± 1.70° (PCL86_2_f, *p* = 0.64) with the oxygen plasma treatment for 8 s with 0.1 bar and 40 W. The lowest contact-angle values were observed for treatment c for the design variant PCL86_2 (12.28 ± 0.37°; significantly reduced compared to the untreated specimens, *p* = 0.08). Design variant 1 showed a much higher value after the same plasma-treatment procedure. Treatments b, f and h showed mean values below 50° (Figure 4) for both design variants and were therefore chosen for further cell-culture experiments.

### 3.3. Influence of 3D-Printed Specimens on Cellular Growth

In the L929 cell culture, significantly lower cell viability in design variant 1 compared to the control was observed after 24 h. After 3 days, both PCL86 variants showed significantly lower cell viability compared to the control (only cells without specimen). After 7 days, no significant differences between the control and both design variants were detected (Figure 5).

### 3.4. Influence of Design Variant and Plasma Treatment of Specimens on Viability of Hbm-MSCs

The differences in plasma treatment had a greater impact on the cell viability than the design variation. In particular, treatment b had a negative influence on both design variants (Figure 6A). A significant improvement on hbm-MSC cell viability was not observed in any of the treatment procedures compared to the untreated specimen. However, confocal microscopic images of the cells on the different surfaces showed a better attachment on the plasma-treated variants compared to the untreated specimens (Figure 6B).

The coating of the implants with fibronectin (PCL86_2_f_fib) improved the hbm-MSC cell viability significantly compared to the only-plasma-treated (PCL86_2_f) and control specimens without any treatment (PCL86_2, Figure 7A). The collagen coating showed a trend towards better cell viability after 24 h (*p* = 0.06) and three days (*p* = 0.072), as well as compared to the untreated group. No differences were observed between the coating groups.

In the descriptive evaluation of the cellular adhesion on the implant surface, pronounced differences were observed, especially after seven days. By contrast, the cells on the untreated specimens did not show any particular direction in their cytoplasms and the plasma treatment increased the alignment of the spindle-shaped cells. The coating further improved this alignment. More prominent spreading between the pores of the construct was observed on the collagen-coated specimens (PCL86_2_f_coll, Figure 7B).

## 4. Discussion

The aim of this study was the development and in vitro characterization of 3D-printed PCL-based scaffolds as possible candidates for tendon-implant applications. Three different types of scaffold (PCL45, PCL86_1 and PCL86_2) were produced and evaluated further.

Compared to the previously used electrospun fiber mats [17,28,29], all the tested 3D-printed specimens were superior in maximum force at break and stiffness, and showed decreased elongation at break. This is desirable for their later use as tendon-implant materials. Therefore, the first hypothesis, that the 3D-printed scaffolds would be superior to the electrospun material, was confirmed, supporting the work of other authors, who contended that electrospun fiber mats did not provide high mechanical strength [31]. The highest strengths were achieved by the PCL45 scaffolds. The slightly lower values of the PCL86 scaffolds might have been due to the higher printing temperatures, which were thought to have an effect on the microstructural properties, thereby harming the mechanical strength [31,32,33]. The slightly thicker struts of the PCL45 scaffolds might also have had an effect. The adaption of the inlay design could have increased the mechanical strength by increasing the interconnectivity of the PCL struts. Although the PCL45 had slightly superior mechanical properties, we chose the PCL86 for further evaluation due to its medical-grade purity and, therefore, better cytocompatibility [6,34,35].

For the general testing of the materials’ cytocompatibility and the possible influence of the design variants of the specimens, in a first step, L929 fibroblasts in direct contact with the unmodified material were used for the cell-viability tests. After 24 h, the growth of the cells in contact with design variant 2 was slightly better compared to design variant 1. This might be explained by the fact that more cells were able to adhere on the specimens with diagonal interlayers, which reduced the pore diameter. After three days, both variants showed lower cell viability compared to the control. However, this effect was no longer detectable after 7 days. With the exception of the 3-day time point, viability of over 70% viability was achieved for the untreated cell control; this value is thought to be a basic requirement for cytocompatibility and further use in the body [36,37]. Due to the direct interactions of degradable materials with cells and possible high concentrations of degradation products in direct contact, viability of 70% is not as strict a value as that required for inert materials regarding biocompatibility. The direct contact methods in the ISO standards were developed for permanent materials. For degradable materials, the protocols should be thoroughly assessed and adapted [38]. Although the slightly acidic degradation products of PCL might cause local acidosis, it has to be considered that the degradation products of slow degrading materials might not accumulate in vivo. This might be due to constant removal via respective processes and, therefore, may not exert negative effects on the surrounding tissue [39].

In our study, different plasma modifications were performed on the surfaces of the PCL-specimens to enhance their hydrophilicity, and thereby, improve their cytocompatibility, which was further tested with the hbm-MSCs. Contact angles under 90° are defined as hydrophilic in technical matters [40]. In our study, the enhancement of the hydrophilicity, understood as the reduction in the contact angle, was highest and most homogeneous for both design variants with the following parameters: b (1 min, 0.3 mbar, 30 W; contact angles: d1: ~45°, d2: ~50°), f (8 min, 0.1 mbar, 40 W; contact angles: d1: ~16°, d2: ~26°) and h (10 min, 0.1 mbar, 40 W; contact angles: d1: ~36°, d2: ~38°). The plasma modification of polymers is a well-known method to introduce functional groups on polymer surfaces [22,41,42]. However, it is crucial to determine the optimal parameters for each specimen, which enhance the hydrophilicity but do not change the morphology of the scaffolds. We already had to exclude some parameters after the first treatment because of obvious changes in morphology. Two of the excluded plasma treatments had the highest values for time and energy, which might explain these morphological changes (fusion, color changes, shrinking). Unfortunately, it remains unclear why d and g were negatively affected, as they had lower parameters (d: 3 min, 0.1 mbar, 30 W; g: 10 min, 0.1 mbar, 30 W) compared to the plasma modification h (10 min, 0.1 mbar, 40 W), which showed positive results (no macroscopic changes and reduced contact angle).

For further specification, human-bone-marrow-derived mesenchymal stem cells were chosen due to their regenerative potential [43]. They can directly differentiate into mesenchymal cell lineages [20] and are easy to handle. The comparison of the chosen plasma-modified specimens (b, f, h) showed reduced cellular viability for b (which had the lowest hydrophilicity). Modifications f (highest hydrophilicity) and h (hydrophilicity between b and f) showed similar values to the untreated specimens, which was surprising as we expected to observe better cell-surface-adhesion properties. However, the cellular growth on the surface of specimens, which was evaluated descriptively through confocal microscopy, was more pronounced in the plasma-treated specimens b and f compared to the native control, which showed nearly no growth on the direct surface. Plasma modification h showed the lowest cell growth on the specimens´ surfaces compared to the other plasma-modified specimens. Other authors were able to show that a contact angle of around 70° is the most compatible with cells [44]. Plasma modification b showed a contact angle slightly below 50°. However, since the other two variants also showed even lower contact angles, it remains unclear why variant b had such low viability values. Other authors used chemical techniques to hydrolyze 3D-printed PCL scaffolds. Kosik-Kosiol et al. produced an increase in surface roughness with acetone and sodium hydroxide treatment, which significantly reduced the water contact angle from 77° ± 4 for the untreated control to 62° ± 2 for the treatment-group and, therefore, increased the growth of human mesenchymal stem cells on the implant surface [45]. Park et al. showed that the implementation of PEG molecules during the 3D-printing process and subsequent extraction by water enabled the creation of micropores on the surface, which led to enhanced surface roughness and cell growth [46]. In addition to the fact that plasma treatment alone was not able to significantly enhance the cell viability of hbm-MSCs, a further important aspect is that hydrophilicity induced by plasma treatment is of short duration [47]. Therefore, it is necessary to improve the attraction for cells for later clinical applications. In the present study, we combined plasma treatment with collagen I or fibronectin coatings. The cell-culture tests demonstrated that the plasma-modified specimens with subsequent fibronectin coating showed significantly enhanced cell viability compared to the unmodified or plasma-modified scaffolds. The collagen I coating showed a trend towards increased cell viability, although this was not significant. However, according to the confocal microscopy, the collagen-I-coated scaffolds showed significant cell growth on the surfaces, as well as interconnective growth. This might have been due to the high viscosity of the collagen I, resulting in thin collagen gels over the pores, which enabled cells to not only grow on the specimens´ surface but also in the interspaces. Large pores seem to be one of the greatest problems for cell growth on scaffolds [20]. Many research groups tried to overcome this problem by bridging the pores with other polymers using different manufacturing techniques, such as electrospinning [48]. Bridging the pores using proteins is a way of combining the positive effects of enhanced interconnectivity and biological surface activation. Other authors found similarly positive results from protein coating. Arredondo et al. investigated commercial scaffolds for meniscus replacement and showed that the functionalization of a PCL–polyurethane surface with fibronectin enhanced the attachment of mesenchymal stem cells due to their integrin α5beta1, which can directly bind to fibronectin [49]. It is well known that the α5β1 integrin plays an important role in essential biochemical and biomechanical signal pathways for the induction of cell adhesion and proliferation [50]. He et al. also examined 3D-printed polycaprolactone scaffolds. While the basic material (PLCL (PLLA:PCL, 50:50) was slightly different, as was the manufacturing process (the authors used solvent-based low-temperature deposition manufacturing), they also used collagen I, which increased the cell viability and the cell morphology on the scaffolds´ surfaces [51]. All these findings conform with our results, as we also showed that additional protein coating can enhance cytocompatibility using hbm-MSCs. Therefore, we recommend an additional coating with fibronectin or collagen I on plasma-modified specimens.

Even though cytocompatibility could be enhanced, it must be concluded that both types of PCL86 specimens still showed minor mechanical strengths compared to commercial augmentation patches for tendon repair [52]. Further experiments need to be performed in order to examine, whether up-scaled specimens for clinical application, using a separated multi-layered design for increasing mechanical properties [32], can reduce the risk of tendon re-ruptures, which mostly occur in the residual degenerated tendon material. Another way of addressing this problem was suggested by Dong et al., who showed that blending PCL with aluminiumoxide whiskers can increase the mechanical strength of electrospun PCL scaffolds [53].

## 5. Conclusions

We were able to show that the use of 3D-printed scaffolds is a suitable starting point for the development of tendon implants with improved mechanical properties and biocompatibility. This can be achieved through design adaptions and surface modifications. Although the mechanical properties and cell compatibility of the scaffolds produced here still require alterations in order to be used in tendon ruptures, we are optimistic that a suitable implant can be designed through further adaptations.

## Figures and Tables

**Figure 1 jfb-13-00160-f001:**
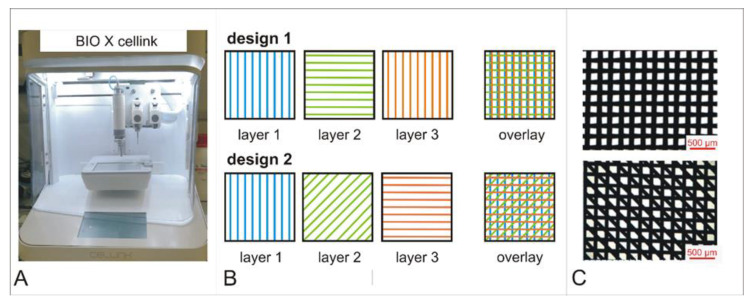
BioX Cellink 3D printer (**A**); scheme of the two printed design variants (**B**) and microscopic picture of final constructs. Scale bar = 500 µm (**C**).

**Figure 2 jfb-13-00160-f002:**
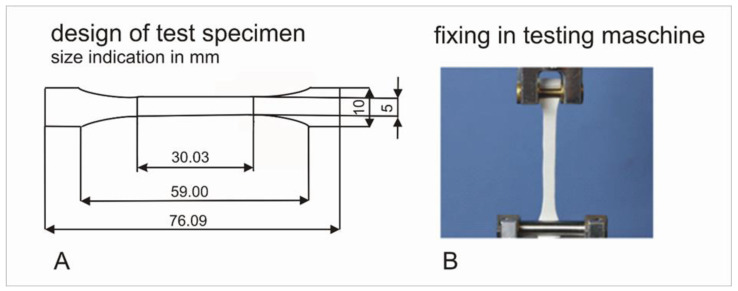
Design of test specimen according to DIN EN 5272 (**A**); implementation of test specimen in the universal testing machine for tensile testing (**B**).

**Figure 3 jfb-13-00160-f003:**
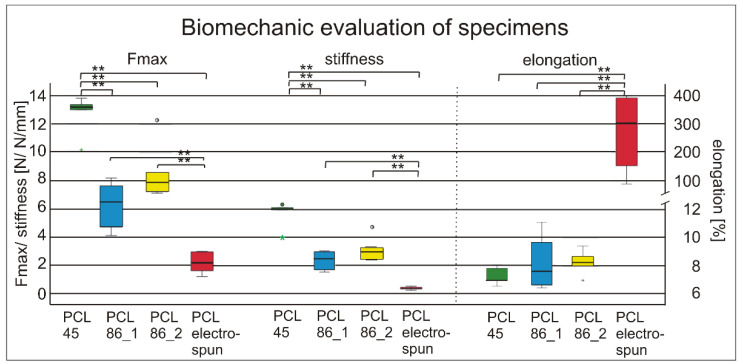
Maximum forces, stiffness and elongation at break differed between different PCL bulk materials, whereas printing pattern (PCL86_1; PCL86_2) showed only minor effects. * = *p* < 0.05; ** = *p* < 0.01; left axis for N and N/mm, right axis for elongation in %.

**Figure 4 jfb-13-00160-f004:**
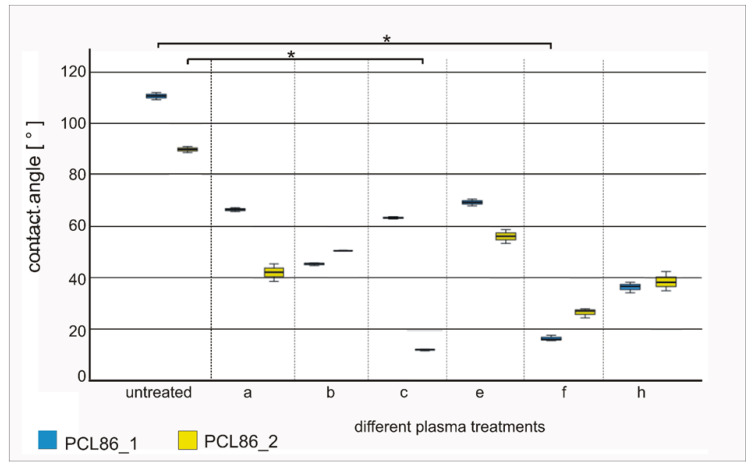
Contact angles of different plasma-treated specimens compared to native specimens of both design variants (design 1 = blue and design 2 = yellow); plasma treatments b, f and h showed lowest and for both design variants most equal contact angles; * marks significant differences (*p* < 0.05).

**Figure 5 jfb-13-00160-f005:**
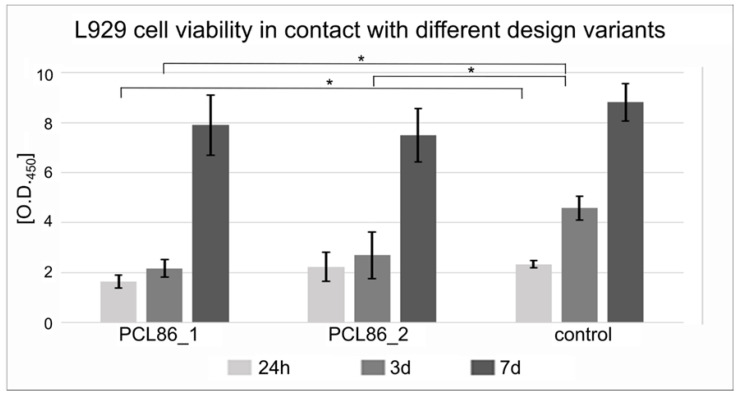
Design variant 1 showed lower cell viability of L929 cells after 24 h and both design variants after 3 d compared to the control group. After 7 d, no statistically significant differences were observed, although both variants showed slightly lower values than the control; * marks significant differences (*p* < 0.05).

**Figure 6 jfb-13-00160-f006:**
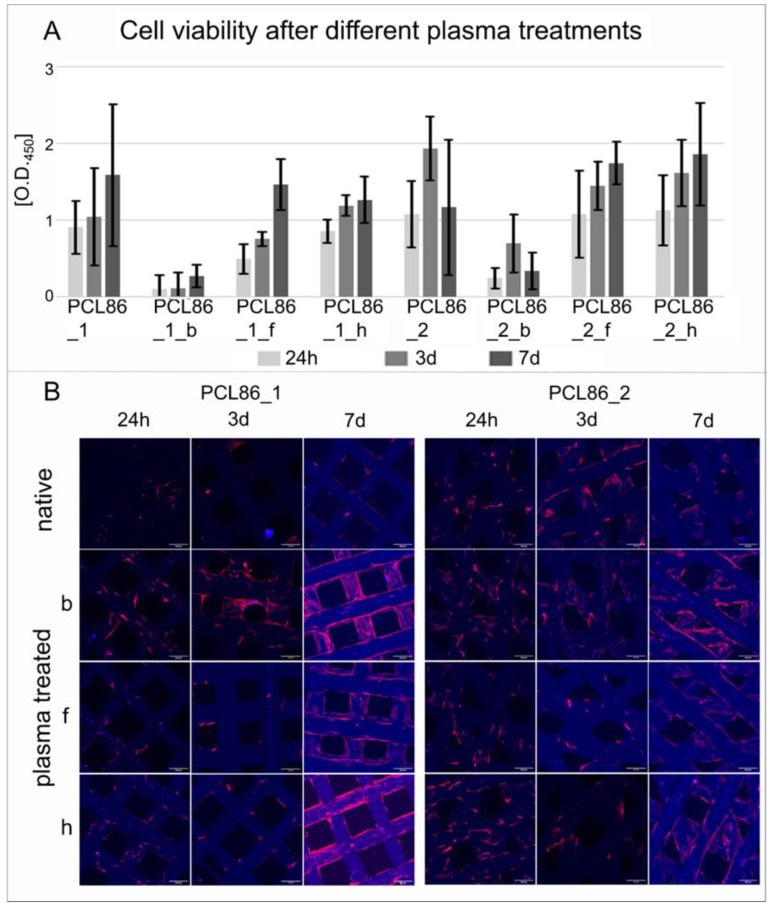
Cell-viability testing of hbm-MSCs showed no significant differences for all groups (**A**); confocal microscopic pictures of fluorescent stained specimens (red = actine skeletons of the cells; blue = cell nucleus) showed slightly more cell growth on the plasma-treated specimens compared to untreated specimens (**B**). 3D-printed specimens also showed some blue fluorescence. Scale bar = 100 µm.

**Figure 7 jfb-13-00160-f007:**
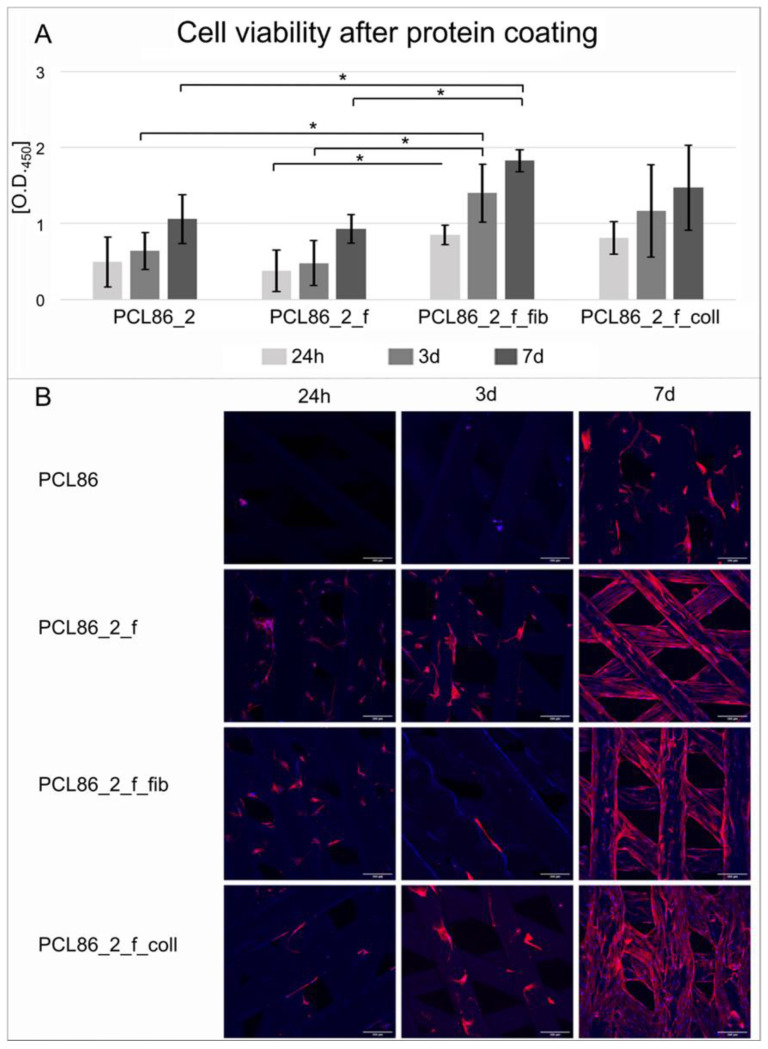
Cell-viability tests showed increased viability of cells that were incubated with coated specimens (PCL86_2_f_fib/coll) compared to only-plasma-treated (PCL86_2_f) and untreated (PCL_2) materials; * marks significant differences (*p* < 0.05) (**A**). Confocal microscopic pictures of fluorescent stained specimens showed more aligned cells on all treated specimens compared to the control, especially after 7 d. Collagen coating (PCL86_f_coll) showed more cells beginning to bridge pores (**B**). White scale bar = 100 µm.

**Table 1 jfb-13-00160-t001:** Specifications of different used polycaprolactone starting materials.

Distributor	Purity	Molecular Weight (g/mol) (Manufacturer Information)	Molecular Weight (g/mol) (Own Measurements)	Further Name
Sigma, Darmstadt, Germany	technical	45.000	34.000	PCL45
ITV, Denkendorf, Germany	medical	-	86.000	PCL86

**Table 2 jfb-13-00160-t002:** Different plasma treatments used for surface hydrophilization.

Group	Time (min)	Pressure (mbar)	Energy (W)
a	1	0.1	30
b	1	0.3	30
c	2	0.1	30
d	3	0.1	30
e	3	0.3	40
f	8	0.1	40
g	10	0.1	30
h	10	0.1	40
i	10	0.1	50
k	10	0.1	40

**Table 3 jfb-13-00160-t003:** Different cell-culture tests performed for cytocompatibility testing.

Scientific Question	Cell Type	Cell Viability Test	Microscopy
Influence of design variants PCL86_1 vs. PCL86_2	L929	24 h (n = 4) 3 d (n = 4) 7 d (n = 4)	-
Influence of design variants PCL86_1 vs. PCL86_2 and plasma treatments a/b/c/e/f/h	hbm-MSC	24 h (n = 6) 3 d (n = 5) 7 d (n= 4)	24 h (n = 1) 3 d (n = 1) 7 d (n = 4)
Influence of fibronectin and collagen I coating on plasma-treated (f) PCL86_2 variant	hbm-MSC	24 h (n = 6) 3 d (n = 5) 7 d (n = 4)	24 h (n = 1) 3 d (n = 1) 7 d (n = 4)

## Data Availability

The datasets used and/or analyzed during the current study are available from the corresponding author on reasonable request.

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
