# Peer review of "Polycaprolactone-Based 3D-Printed Scaffolds as Potential Implant Materials for Tendon-Defect Repair"

_jfb, 2022, doi:10.3390/jfb13040160_

Round 1
Reviewer 1 Report
Please follow the recommendations given to revise the manuscript

Author Response
Dear reviewer,
thank you for the valuable remarks to improve the manuscript. The point by point response to your comments is attached as word file.
Kind regards
Janin Reifenrath

Reviewer 2 Report
In this manuscript, the authors present modified 3D-printed PCL scaffolds for tendon repair. Although some studies using 3D-printed PCL scaffolds have already been published, additional studies contributing to increased acceptance of this material in regenerative medicine are more than welcome. Overall, the manuscript is well written and the results of this work are of interest to readers. However, some issues should be addressed before publication:
1) Since the authors use different approaches for tailored physicochemical and biological properties of different PCL, some additional surface characterization tests would increase the scientific value of the manuscript. For example:
· FT-IR spectroscopy to confirm biomimetic coatings and the formation of functional groups in plasma-treated PCLs compared to control;
· SEM (with and without seeded cells) to evaluate the morphology of different scaffold designs (and their effects on seeded cells).
2) Table 3 should be revised to improve readability.
3) alpha5beta1 in Lines 418-419 should be written with Greek letters.
Author Response
Dear reviewer,
thank you for your valuable remarks to improve the manuscript. Attached you find the point-by-point response to your comments.
Kind regards
Janin Reifenrath

Reviewer 3 Report
The authors systematically studied the 3D printed pattens, surface hydrophilization with plasma treatment and molecular weight of PCL as a potential implant material for tendon repair.
1) PCL as an implant material for tendon repair has been well documented, the novelty is low.
2) In fact, the different pattens of printed structure and electropsun structures can be treated as different pore size and porosity effect on tendon repair, these have been intensively studied in the literature, the authors should comment in the introduction.
3) further evaluations are required such as relavant factors and gene expressions should be given apart from cell experiments.
Author Response
Dear reviewer, thanks for your remarks. We tried to thoroughly answer your comments
The authors systematically studied the 3D printed pattens, surface hydrophilization with plasma treatment and molecular weight of PCL as a potential implant material for tendon repair.
- PCL as an implant material for tendon repair has been well documented, the novelty is low.
- In fact, the different pattens of printed structure and electropsun structures can be treated as different pore size and porosity effect on tendon repair, these have been intensively studied in the literature, the authors should comment in the introduction.
We would like to comment on these two points together.
While it is correct, that studies on PCL for tendon application already exist, there is currently no approved PCL based tendon implant on the market. Furthermore, the goal of our approach is to combine the material with sufficient mechanical and biocompatibility properties. PCL as FDA approved material induced e.g. massive foreign body reactions in vivo when used as electrospun-fibre mat with very small fibre diameter (Willbold et al., 2020). Therefore, more research is needed, when design varies. We added this to the introduction. Additional literature is marked yellow in the reference list.
“In the literature it has been already studied as tendon implant material with different electrospun or woven design variants [10, 11, 12, 13], but until now there is no commercially available PCL-construct for tendon ruptures [14]. PCL as FDA approved material induced e.g. massive foreign body reactions in vivo when used as electrospun-fibre mat with very small fibre diameter [15]. Therefore, more research is needed, when design varies.”
- further evaluations are required such as relavant factors and gene expressions should be given apart from cell experiments.
It is an interesting and important approach to include gene expression analysis. Our study focused on mechanical properties, surface functionalization possibilities and biocompatibility aspects. The chosen time points suited this focus while gene expression analyses – in our opinion - would be more interesting for later time points (differentiation of stem cells). Therefore, future studies might include such techniques.
Reference:
Willbold, E., Wellmann, M., Welke, B., Angrisani, N., Gniesmer, S., Kampmann, A., Hoffmann, A., de Cassan, D., Menzel, H., Hoheisel, A. L., Glasmacher, B., & Reifenrath, J. (2020). Possibilities and limitations of electrospun chitosan-coated polycaprolactone grafts for rotator cuff tear repair. Journal of tissue engineering and regenerative medicine, 14(1), 186–197. https://doi.org/10.1002/term.2985
Round 2
Reviewer 1 Report
The authors have carefully revised the manuscript according to the comments and the quality of the manuscript has been greatly improved, so I recommend it for publicationAuthor Response
The authors thank the reviewer for improving the manuscript.
Reviewer 2 Report
The authors satisfactory justified my comments.
Author Response
The authors thank the reviewer for improving the manuscript.
Reviewer 3 Report
Evaluation in depth is required such as growth factors etc.
Author Response
The authors thank the reviewer for improving the manuscript. A careful spell check is perfomed additionally. In future studies, evaluation of growths factors are interested to implement.